# A Guide to Perform 3D Histology of Biological Tissues with Fluorescence Microscopy

**DOI:** 10.3390/ijms24076747

**Published:** 2023-04-04

**Authors:** Annunziatina Laurino, Alessandra Franceschini, Luca Pesce, Lorenzo Cinci, Alberto Montalbano, Giacomo Mazzamuto, Giuseppe Sancataldo, Gabriella Nesi, Irene Costantini, Ludovico Silvestri, Francesco Saverio Pavone

**Affiliations:** 1European Laboratory for Non-linear Spectroscopy, LENS, 50019 Sesto Fiorentino, Italy; annunziatina.laurino@unifi.it (A.L.); franceschini@lens.unifi.it (A.F.); pesce@lens.unifi.it (L.P.); alberto.montalbano@unifi.it (A.M.); giacomo.mazzamuto@ino.cnr.it (G.M.);; 2Department of Physics and Astronomy, University of Florence, 50019 Florence, Italy; 3Department of Experimental and Clinical Biomedical Sciences, Radiodiagnostic Unit n. 2, Careggi University Hospital, 50134 Florence, Italy; lorenzo.cinci@unifi.it; 4Department of Neurofarba Section of Pharmacology and Toxicology, University of Florence, 50139 Florence, Italy; 5National Research Council—National Institute of Optics (CNR-INO), 50125 Sesto Fiorentino, Italy; 6Department of Health Sciences, University of Florence, 50139 Florence, Italy; 7Department of Biology, University of Florence, 50019 Florence, Italy

**Keywords:** 3D histology, clearing methods, H&E staining, Light Sheet Microscopy, Two Photon Fluorescence Microscopy

## Abstract

The analysis of histological alterations in all types of tissue is of primary importance in pathology for highly accurate and robust diagnosis. Recent advances in tissue clearing and fluorescence microscopy made the study of the anatomy of biological tissue possible in three dimensions. The combination of these techniques with classical hematoxylin and eosin (H&E) staining has led to the birth of three-dimensional (3D) histology. Here, we present an overview of the state-of-the-art methods, highlighting the optimal combinations of different clearing methods and advanced fluorescence microscopy techniques for the investigation of all types of biological tissues. We employed fluorescence nuclear and eosin Y staining that enabled us to obtain hematoxylin and eosin pseudo-coloring comparable with the gold standard H&E analysis. The computational reconstructions obtained with 3D optical imaging can be analyzed by a pathologist without any specific training in volumetric microscopy, paving the way for new biomedical applications in clinical pathology.

## 1. Introduction

Three-dimensional (3D) reconstruction at cellular or sub-cellular resolutions of intact biological specimens could offer several advantages in both clinical and pre-clinical studies, for example, in cancer diagnosis, by helping to accurately determine the disease stage [1], as recently reported [2], or to study anatomical alterations with cell-type specificity on whole animal organs after the application of specific stimuli or treatments [3]. Using classical histological techniques, such as hematoxylin and eosin (H&E) staining and immunohistochemistry, tissues need to be fixed or frozen, sectioned, mounted, and then stained before being acquired. Although progress has been made to automate this experimental pipeline, slide-based protocols are time-consuming and labor-intensive, presenting several intrinsic limiting factors due to the bidimensional nature. The most common drawbacks include: (i) sampling errors or inaccurate qualifications, since the entire tissue is not analyzed; (ii) difficult assessment of dimensions and morphology due to visual artifacts; (iii) difficulty in establishing the correct orientation of the sample or the edges of a lesion; or (iv) destruction of the tissue: the sample is inevitably subjected to a cutting process that hampers further investigation. In addition to these drawbacks, tissue processing for histology also requires a large amount of skilled labor and equipment [4]. 

Recently, technological developments in tissue clearing and fluorescence microscopy opened the possibility of staining and imaging large specimens, even whole organs [1,5,6,7]. The principle of tissue clearing relies on the homogenization of the refractive index (RI) inside and outside the sample, thus allowing good penetration of the dyes and optical sectioning of intact samples. Each clearing method has advantages and limitations according to the tissue to be analyzed (e.g., size, fixation, conservation) and the goal to be achieved (e.g., resolution, acquisition speed) [8,9,10]. 

In this work, due to their spectral properties and compatibility with widespread commercial and custom microscopes, we coupled Eosin/DAPI or Eosin/SYTOX Blue staining to different clearing techniques such as CLARITY, SWITCH, MAP, and iDISCO [11,12,13,14], trying to highlight the advantages and limitations of each approach in order to achieve the cellular and/or subcellular reconstruction of intact mouse organs or large human specimens. We applied our protocols to formalin-fixed mouse kidney, liver, bladder, skeletal muscle, spinal cord, small intestine, and brain samples, as well as to formalin-fixed human hippocampus and bladder samples and to a formalin-fixed paraffin-embedded (FFPE) tumor xenograft sample. To verify the compatibility of 3D histology with the classical histological approach, we converted the fluorescent images into pseudo-colored images suitable for pathologic evaluation (Figure 1). This work allowed us to verify the characteristics of each clearing technique and to suggest the best combination with the appropriate imaging technique.

In particular, we found that SWITCH is preferred for clearing human brain samples, especially when the fixation conditions are not optimal, and MAP works better than CLARITY with FFPE tissues. However, the staining is compatible with all the clearing methods tested (CLARITY, SWITCH, MAP, iDISCO) and with two different microscopy techniques, confirming, where present, previous literature data.

### Literature Review

H&E is the gold standard in pathologic evaluation, and it is routinely used in critical practice to provide information about the anatomical structure of biological tissue.

With the aim of using the best match of clearing protocols and imaging techniques for the 3D histology of large samples, several combinations of fluorescent dyes analog to H&E have been proposed. For the majority of the combinations proposed, the process involves a series of steps including clearing, staining, RI-matching, imaging, and post-processing.

The storage of biological material is the first thing to be considered when choosing the most suitable tissue clearing protocol. Undoubtedly, fresh tissues represent the optimal condition, but they are often not available. Many tissue samples are stored for a long time in formalin or are FFPE to avoid cell and tissue degradation for further analysis at any time. However, the compatibility of 3D histology was demonstrated with both fresh [15], formalin-fixed [16], and FFPE tissues [17,18,19,20]. In particular, iDISCO, or its variants, demonstrated high compatibility with both fresh and FFPE human tissues [19,21,22,23]; other methodologies working on fresh/formalin-fixed/FFPE include CUBIC [17,20], ACT-PRESTO [16], BABB [24], X-CLARITY [15], PACT [25,26], FACT [27,28], and MASH [29,30]. 

Labeling is the second important aspect of tissue preparation for 3D histology. To mimic classical H&E, i.e., nuclear and cytoplasmic staining, several combinations of inexpensive (compared to antibodies) small-molecular probes have been proposed. In particular, for fresh and uncut biopsies, Elfer et al. [31] validated Draq5/Eosin staining, which was applied by Glaser et al. [15] to X-CLARITY-processed samples. A similar approach, with a combination of 4′,6-diamidino-2-phenylindole (DAPI)/Eosin, was provided by Olson et al. [24] to be applied to both fresh and 24 months-fixed samples of 1-mm thickness. Other commonly used nucleic acid stains are SYTO 16 [16,20], SYTOX Green [17], and TO-PRO-3 [17,21,23]. Eosin is the most frequently reported cytoplasmic stain. However, Alexa 647 [17], Rhodamine [18], and autofluorescence enhancement with 5-sulfosalicylic acid dihydrate are also suitable for 3D histology [19]. 

As regards imaging, several fluorescence microscopy methods can be used to obtain volumetric acquisition of whole samples. Confocal and multiphoton microscopy offer high contrast and resolution [16,24], although the slow acquisition speed represents an intrinsic limitation for the acquisition of large-volume tissues. On the other hand, Light Sheet Microscopy (LSM) allows rapid 3D acquisition of cleared samples, and it is compatible with several H&E-analog staining methods [15,17,21]. Once the acquisition is finished, images are generally post-processed to provide 3D volumetric reconstructions and, eventually, pseudo-colored to mimic classical H&E colors [17,24]. 

Due to the variability of biological applications, a standard technique to perform 3D histology is still not present; on the contrary, as described above, various solutions have been proposed. In the next sections, we present some applications we performed with the most widespread clearing techniques published in the last few years in order to provide a guide of 3D histology on different kinds of samples.

## 2. Results

### 2.1. Eosin/SYTOX Blue Staining on CLARITY-Processed Samples

CLARITY [11], introduced in 2013, was the first tissue transformation technique [3,6]. Such methodology uses specific crosslinkers (i.e., acrylamide and bis-acrylamide) to stabilize the endogenous proteins and nucleic acids of tissues, forming a heat- and chemical-resistant hydrogel compatible with lipid extraction. The removal of lipids allows the reduction in the refractive index of the tissue, which can be matched with aqueous solutions, e.g., PBS with Thiodiethanol [32], and the homogeneous labeling of the sample, since the hydrogel mesh is easily permeable to macromolecules. 

We used the passive CLARITY/TDE [32] technique followed by Eosin/SYTOX Blue staining to study intact mouse organs (Figure 1(b1)). The 3D reconstruction of the whole samples was obtained using a custom LSM [33]. After LSM reconstruction, the data were converted into pseudo-colored images using a transformation of the color space, after normalization, according to the formulas provided by Torres et al. [34]. We performed the protocol on a whole mouse bladder and portions of skeletal muscle, spinal cord, kidney, and liver. 

For hollow organs such as the bladder (Figure 2a), thin (~1.5 mm) specimens such as the spinal cord (Figure 2b), and the skeletal muscle (Figure 2c), the procedure led to optimal transparency and staining. The cytoarchitectural staining patterns are comparable with those observed with classical H&E in bright field microscopy. In the bladder, the urothelium (U) is easily identifiable (Figure 2a), while in the spinal cord (Figure 2b), the ependymal canal (EC) is recognizable. In the skeletal muscle sample (Figure 2c), the typical position of the nuclei (Nu) is apparent at the periphery of the myofiber. In solid organs such as the kidney and liver, Eosin penetration and SYTOX Blue penetration were successfully achieved throughout the sample (Figure 2d,e), although striping artifacts are evident in Eosin staining. Indeed, the typical functional unit of the kidney, the glomerulus (G), and the high acidophilic cytoplasm of hepatocytes in the liver are visible.

### 2.2. Eosin/DAPI Staining on SWITCH-Processed Sample

SWITCH [12] was selected to treat human samples, in particular, a ~500 µm-thick slice of a human hippocampus, for its compatibility with non-controlled fixation conditions [35,36] (Figure 1(b2)). We chose a thickness of 500 µm since our previous works demonstrated that this is the best dimension to be used to achieve optimal transparency with human brain samples. Indeed, the tissue autofluorescence, highly present in this kind of tissue, prevents the light from penetrating deeper into the sample, lowering the signal-to-background ratio and, thus, limiting the thickness of the section where it is possible to observe homogeneous staining [8,10,35,36]. Nevertheless, SWITCH is also suitable with mouse organ clearing [12]; therefore, we applied it also to a segment of a mouse small intestine and to a section of a mouse kidney. The SWITCH clearing process was followed by Eosin/DAPI staining and coupled with the TDE index-matching [35,36]. Samples were then imaged with Two Photon Fluorescence Microscopy (TPFM). As shown in Figure 3, after pseudo-coloration, a staining very similar to that of classical H&E is obtained. In particular, in the small intestine, the columnar epithelium (E) and goblet cells (GC) are evident (Figure 3a and Appendix A). In the human hippocampus, the large, round nucleus of neurons (N) with the nucleolus, and that of glial cells (Gl), which is smaller and with intensely colored heterochromatin (Figure 3b), were observed. In the kidney, glomeruli (G) and tubules (T) are clearly distinguished (Figure 3c).

### 2.3. Eosin/SYTOX Blue Staining on MAP-Processed Sample

Three-dimensional reconstruction of the xenograft tumor sample requires making a chemical- and heat-resistant gel/tissue hybrid to perform an efficient clearing process. To this aim, we tested both the CLARITY and SWITCH methods, but we did not reach optimal transparency. Indeed, fibrotic components in the tumor tissue make the clearing process challenging. To tackle this issue, we applied MAP [13] on a ~0.5 × 0.5 × 1 cm^3^ tumor block, a tissue transformation protocol that exploits the swelling properties of the hydrogel backbone to perform the clearing process. In MAP, a high concentration of acrylamide during the hydrogel-tissue hybridization step prevents protein–protein cross-linking caused by formaldehyde reaction. Such reduction in intra- and inter-protein crosslinking generates an efficient dissociation of protein complexes and improves tissue expansion. Furthermore, the ability to preserve the epitopes after the homogenization process allows for retaining and staining multiple targets after expansion. The robustness of the hydrogel, due to the high-density mesh, and protein retention ability lead us to prefer MAP instead of the ExM classical protocol for eosin staining [37,38]. The high water absorption reduces the refractive index (the specimen is ~99% water), leading to an efficient clearing of the xenograft tumor. The MAP process was combined with Eosin Y and SYTOX Blue staining. Images were acquired with the custom LSM (Figure 4a) [33]. With this protocol, despite the high cell density, we obtained a complete reconstruction of the sample with a subcellular resolution (~250 nm), reducing stripe artifacts. After pseudo-coloration, the staining was comparable to that of classical H&E. In particular, it is possible to note the granular aspect of chromatin resembling the “salt and pepper chromatin” (SPCh), presumably due to an apoptotic phenomenon. Being able to identify apoptotic cells is important, for example, to evaluate the effectiveness of anticancer therapies [39]. We also applied the MAP technique combined to Eosin/SYTOX Blue staining to perform a complete reconstruction of a mouse brain section (Figure 1(b3)). Images were acquired with LSM (Figure 4b) [33]. The lateral ventricle (LV) and, in the zoom-in images, the round nucleus of neurons (N) with nucleolus are clearly visible.

### 2.4. Eosin/SYTOX Blue Staining on iDISCO-Processed Sample

Finally, we tested the iDISCO technique (Figure 1(b4)). This technique is based on tissue dehydration and RI-matching processes. Among the abovementioned tissue clearing protocols, iDISCO is largely used, since it is a simple and rapid method to make all types of specimens extremely transparent. Indeed, clearing and Eosin/SYTOX Blue labelling steps last only three days [14,40,41,42,43]. For this reason, organic solvents are the most commonly employed techniques to clear tissues [6], and several authors have demonstrated their compatibility with 3D histology [19,21,22,23]. Despite the advantages of rapid “tissue-transparentizing”, the use of many toxic and dangerous agents and the scarce availability of appropriate immersion lenses for imaging led researchers to also follow/adopt alternative clearing protocols.

Here, we combined the iDISCO protocol with Eosin/SYTOX Blue Staining on a small sample of a human bladder (~0.5 × 0.5 × 2 cm^3^). Images were acquired with the custom LSM [33] (Figure 5 and Appendix A). Additionally, with this technique, after pseudo-coloration, the staining is comparable to that of classical H&E (Figure 5). In the 3D reconstruction, portions of Adipose tissue (A) and of Muscularis propria (M) are evident. In the zoomed-in images, the urothelium (U) is clearly visible (Figure 5). 

## 3. Discussion

Classical histological techniques, such as H&E staining, although universally adopted in almost all clinical and preclinical laboratories throughout the world, suffer from several intrinsic limitations, mainly due to the restriction to 2D analysis. To address this issue, several authors proposed to couple a staining that mimics the classical H&E with recently developed clearing techniques [6], to obtain a complete 3D reconstruction of entire animal organs or large pieces of human samples, giving rise to the so-called 3D histology. A fundamental step to perform 3D histology is to understand the compatibility among the kind of sample (size, species, preservation conditions), the staining (spectral properties, aqueous/organic solutions), and the microscopy (resolution, signal-to-noise ratio, speed, throughput). For this purpose, we analyzed different kinds of samples (in terms of fixation, size, and species) by coupling Eosin/DAPI or Eosin/SYTOX Blue staining with different clearing techniques and by acquiring images with two different microscope setups. Finally, we converted the fluorescent images into pseudo-colored images and asked a histologist to verify the quality of the 3D histology.

We chose CLARITY for mouse whole organs because it works well with fresh fixed tissues. We then imaged it with a custom LSM that acquires images from a large volume of tissue and provides high imaging speed. We obtained successful staining and acquisition for hollow organs such as bladder and thin specimens such as spinal cord and skeletal muscle. However, with solid organs, we observed several striping artifacts in the eosin channel, a typical artifact of LSM in presence of a dense staining and/or imperfect clearing. Moreover, CLARITY is not the optimal clearing technique for human samples. Indeed, only human sections with fine-μm-thickness are cleared with this technique [15]. 

For large human samples, we applied SWITCH, which performs well in the case of non-controlled fixation and conservation conditions and which, in our experience, works well with both human brain specimens [8,35,36] and mouse samples. We observed the samples with TPFM, which allows mesoscopic reconstruction of large, flat sections. In this case, we obtained successful staining for the human brain hippocampus slice, for the segment of a mouse small intestine, and for a slice of a mouse kidney. In particular, the pseudo-colored images allow us to clearly identify single cells and typical structures. However, notwithstanding the high contrast and resolution, this kind of clearing technique coupled with TPFM allows large sections to be analyzed at a lower speed in respect of LSM. Indeed, SWITCH coupled with TPFM is an optimal combination to obtain a subcellular spatial resolution of a small sample. However, being a point-scanning technique, TPFM is not suitable for entire organ reconstruction due to time limitations.

Isotropically expanding biological samples, subcellular or even nanometer resolution is easily achievable. Due to the large size of the sample obtained at the end of the clearing process, MAP-processed tissues were imaged with LSM, which provides significantly higher imaging speed compared to TPFM. The swelling properties of the hydrogel (the specimen is ~99% water at the end of the expansion) reduce the refractive index and lead to less dense eosin staining compared with CLARITY and SWITCH, allowing acquisition with minor striping artifacts. This kind of process was also compatible with a fresh fixed sample (mouse brain). Therefore, with MAP, we were able to stain large volumes of tissue at a higher resolution than that achievable with CLARITY-processed samples and faster with respect to TPFM. The main limitation of this clearing technique is mainly due to the increase in the dimensions due to the expansion of the sample. In this case, the entire organ reconstruction would require too much effort and time, even using LSM.

Since organic solvents are the most frequently used clearing agents, for 3D histology applications, we also evaluated the compatibility of the Eosin/SYTOX Blue Staining and of our custom LSM with the iDISCO protocol. To note, in this case, we dissolved Eosin Y and SYTOX Blue in MeOH, rather than aqueous solutions or TDE. Additionally, with this technique, we achieved successful staining of a small portion of the human bladder. This clearing protocol ensures high transparency of tissues from all species, and a significant advantage is that the clearing procedure is very fast, taking a short time. Anyway, it does not preserve the endogenous fluorescence and also causes sample shrinkage. For this reason, it is not suitable to image fine structures.

Overall, our results demonstrate that there is not a single “killer technique” suited for all samples and all applications. Each of these techniques has its advantages and disadvantages. More realistically, the researcher and the pathologist should choose the clearing, staining, and imaging methodologies that fit best with their respective goals. For instance, sub-cellular imaging of moderately sized specimens will be performed well by MAP preparation and LSM, whilst high-resolution analysis of large tissue blocks would require iDISCO or CLARITY. In this work, we provide an evaluation of different techniques, hoping to guide others in this choice (Table 1). In line with the literature, our results proved that CLARITY clearing methods do not provide high transparency with human samples [15], in contrast with iDISCO, which, coupled with LSM, provides remarkable transparency of large human samples [19,21,23]. Generally, these combinations are used for large-scale reconstructions. However, to observe small details, it is better to use a combination of SWITCH or MAP with TPFM [13,35,36]. MAP is even better to image fine structures. Indeed, it was applied to investigate connections and fine synaptic architectures in the mouse brain [13].

In addition, it must be noted that also some label-free optical methods are emerging as potential tools to explore histopathological tissue samples. These include, e.g., multispectral imaging [44,45,46], Raman spectroscopy [47,48], and non-linear microscopy [49,50]. These approaches are complementary to the 3D imaging presented here, which is based on chemical labeling, in the sense that they are typically slower and thus limited to smaller areas, but provide sample information without the need for any treatment of the samples.

One important and open question concerns the use of 3D histology in clinical settings. Indeed, this technology offers unprecedented opportunities for improved diagnosis [1,2]. However, some improvements are still needed. Concerning sample preparation, processing times have to be reduced as much as possible, maybe with active clearing approaches [51]. On the imaging side, methods to contrast artifacts caused by sample-induced defocus [52] or by striping [53] must be implemented in LSM. Finally, computational tools capable of withstanding the large amount of data produced by 3D microscopy are essential; until TB-sized datasets are manageable by standard facilities, routine use of 3D histology will be impossible. 

Histological analysis of all tissues is highly relevant both in the clinical setting, enabling more accurate and robust diagnosis [1,2], and in basic and preclinical research [3]. This kind of staining which mimics classical H&E coupled with several clearing and imaging techniques has the advantage, compared with antibodies, of being inexpensive and suitable for several tissue types. Reconstructions of intact organs or large human specimens with cellular or even subcellular resolution are achieved with different clearing techniques and microscopes used. In addition, another advantage is that the fluorescence images obtained can be easily converted into pseudo-colored images suitable for pathologists.

## 4. Materials and Methods

### 4.1. Specimens Collection

Adult male and female FosTRAP mice (B6.129(Cg)-Fostm1.1(cre/ERT2)Luo/J × rtB6.Cg-Gt(ROSA)26Sortm9(CAG-tdTomato)Hze/J) were used for this work [54]. They were housed in groups of 3 or 4 with food and water ad libitum and were maintained in a room under controlled light and dark cycle (12/12 h; light starts at 7:00 AM), temperature (22 ± 2 °C), and humidity (55 ± 10%). All experimental procedures were approved by the Italian Ministry of Health (Authorization n. 512-2018_FC). Alternatives to in vivo techniques were not available, but all experiments were conducted according to principles of the 3Rs.

The human healthy tissue was obtained from the Body Donation Program “Donation to Science” of the University of Padova. Prior to death, participants provided their written consent for the use of their entire body for any educational or research purpose in which the anatomy laboratory is involved. The authorization documents (under the form of handwritten testaments) are kept in the files of the Body Donation Program. Upon collection, samples were placed in neutral buffered formalin (pH 7.2–7.4) (Diapath, Martinengo, Italy) and stored at room temperature (RT) until the transformation and clearing process.

### 4.2. Animals Tissue Preparation

Mice were anesthetized with (1.5–2%) isoflurane and perfused transcardially with ice-cold 0.01 M phosphate buffered saline (PBS) solution (pH 7.6) followed by ice-cold 4% paraformaldehyde (PFA). The organs extracted were post-fixed overnight at 4 °C and then cleared using different clearing methods (CLARITY, SWITCH, MAP).

### 4.3. CLARITY Clearing Protocol

Some organs from mice were prepared according to the CLARITY/TDE protocol [11,32]. Immediately after perfusion, biological tissues were post-fixed in PFA overnight at 4 °C. The day after, samples were incubated in a hydrogel solution (containing 10% acrylamide (*v*/*v*), 2.5% bis-acrylamide (*v*/*v*) and 0.25% VA044 (*w*/*v*) in PBS) at 4 °C for 3 days, allowing a sufficient diffusion of the solution into the tissue. Samples were then degassed, replacing oxygen inside the vials with nitrogen, and incubated in a water bath at 37 °C for 3 h in order to initiate polymerization of the hydrogel. After 3 h, embedded organs were placed in a clearing solution (containing 4.4% (*w*/*v*) sodium dodecyl sulfate (SDS) and 1.2% (*w*/*v*) boric acid in ultra-pure water, pH 8.5) at 37 °C. The clearing solution was changed every 2–3 days. Specimens were gently shaken throughout the whole clearing period, which typically takes 3–4 weeks. When the samples appeared sufficiently transparent, they were incubated 1 day in PBS with 0.1 Triton-X (PBST, pH 7.6) and 1 day in PBS (pH 7.6), removing the excess SDS. Samples were stained with an aqueous solution of Eosin Y 0.5% *w*/*v* at RT (Sigma Aldrich, Milano, Italy) for 1 to 24 h, depending on sample size. Then, they were extensively washed with PBS, until the solution was colorless. Finally, murine samples were optically cleared with serial immersions of mixtures containing 20% and 40% 2-2′ Thiodiethanol (TDE) in PBS and 1:5000 *v*/*v* SYTOX Blue, each for 1 day while rotating. The last mixture (40% TDE) was used as an index-matching solution for imaging [55].

### 4.4. Human Brain and Mouse Slices Preparation for SWITCH

Both human brain and mouse samples were embedded in a low melting agarose (4% p/v in 0.01 M PBS) and cut into 450 ± 50 µm coronal sections with a vibratome (Vibratome 1000 Plus, Intracel Ltd., St. Ives, UK). After the cutting, the agarose surrounding each slice was removed. The permeabilization and staining protocols were modified from those of Murray et al., 2015 following the Costantini et al., 2021 [12,13,14,15,16,17,18,19,20,21,22,23,24,25,26,27,28,29,30,31,32,33,34,35] protocol, as described below. Samples were first incubated in the ice-cold SWITCH-OFF solution (4% glutaraldehyde (GA) in PBS 1 X and KHP 0.1 M, titrated with HCl to pH = 3) for 1 day at 4 °C with gentle shaking, then incubated for 1 day in the SWITCH-ON solution (0.5% GA in PBS 1 X, pH = 7.6) for 1 day at 4 °C with gentle shaking. After two washing steps in the PBST solution (PBS with 1% Triton X-100, pH = 7.6) for 4 h at RT, the samples were inactivated with a solution of 4% *w*/*v* acetamide and 4% *w*/*v* glycine with a pH = 9 (overnight incubation at 37 °C). Two washing steps in PBST solution for 4 h at RT were performed before the incubation in the Clearing Solution (200 mM SDS, 20 mM Na2SO3, 20 mM H3BO3, pH = 9) at 70 °C for lipid removal. Incubation time in the clearing solution was adapted, depending on tissue characteristics, until complete transparency was achieved. The samples were then washed with PBS at RT to completely remove SDS. Samples were stained with an aqueous solution of Eosin Y 0.5% *w*/*v* at RT (Sigma Aldrich, Milano, Italy) for 1 h. Then, they were extensively washed with PBS until the solution was colorless. Samples were serially incubated in 2 mL of 20%, 40%, and 68% (*v*/*v*) TDE/PBS, each for 12 h at RT, while gently shaking. After TDE clearing, the nuclei were stained with DAPI 1:5000 *v*/*v* in 68% (*v*/*v*) TDE/PBS (Thermofisher, Rodano, Italy) and imaged.

### 4.5. Deparaffinization from Formalin-Fixed Paraffin-Embedded (FFPE) Tissue

A paraffin-embedded tumor xenograft sample was obtained as described by Bigagli et al. [39]. The sample was deparaffinized by 2 incubation steps with xylene for 1 h at 37 °C and 1 step with xylene for 1 h at RT. The sample was then incubated with a downgraded series of alcohol solutions (100%, 95%, 90%, 80%, and 70% ethanol for 1 h each at RT) and, finally, washed with PBS overnight, as reported by Tanaka et al. [22].

### 4.6. MAP 

For MAP experiments [13], deparaffinized tumor and mouse brain cubes (~0.5 × 0.5 × 1 and ~1 × 1 × 1 cm^3^) were incubated in a solution consisting of 4% paraformaldehyde, 10% acrylamide, 0.05% bisacrylamide, 1.7% sodium acrylate, and 1× PBS in deionized (DI) water for 2 days at 4 °C with gentle shaking to guarantee uniform chemical diffusion and reaction throughout the specimen. The specimen was then polymerized under vacuum at 37 °C for 2 h, and the excess of the hydrogel around the samples was carefully removed. The hydrogel-embedded specimen was incubated in a solution of 200 mM SDS, 200 mM NaCl, and 50 mM Tris in DI water (pH titrated to 9.0) overnight at 37 °C with gentle shaking. Then, the mouse sample was incubated at 70 °C for 8 h, while the tumor specimen was incubated at 70 °C for 2 h and then at 90 °C for 2 h. The denatured specimen was then labeled with Eosin Y 0.5% *w*/*v* at RT (Sigma Aldrich, Milano, Italy) for 24 h and washed 3 times with DI water, 30 min each, for removing the eosin excess. Finally, the stained specimen was incubated in 50 mL DI water with 1:5000 *v*/*v* SYTOX Blue at RT with gentle shaking for 24 h. During DI water incubation, the solution was changed every 3–5 h to reach the complete expansion.

### 4.7. iDISCO Clearing Protocol

A human bladder sample (~0.5 × 0.5 × 2 cm^3^) was prepared according to the iDISCO protocol [14]. Fixed bladder was washed in PBS (pH 7.6) for 1 h twice, then it was treated with increasing concentrations of H_2_O/methanol (MeOH) (20%, 40%, 60%, 80%, and 100%) for the dehydration process, with gentle shaking. The incubation time was 1 h, and it was performed at RT. The day after the dehydration process, the sample was previously stained with 1:100,000 *w*/*v* Eosin Y (Sigma Aldrich, Milano, Italy), dissolved in MeOH 100%. The third day, the sample was washed every hour to remove the excess of eosin. After eosin removal, the somata labeling was performed using 1:5000 *v*/*v* SYTOX Blue (Thermofisher, Rodano, Italy) in MeOH 100%. The last day, the sample was incubated for 3 h at RT, with gentle shaking, in a solution composed by 66% Dichloromethane (DCM) (270997-250 mL, Sigma Aldrich, Milano, Italy) and 33% MeOH 100%, and washed in 100% DCM 20 min twice. Optical clearing was performed by DiBenzyl ether incubation (DBE) (108014-1 Kg, Sigma Aldrich, Milano, Italy).

### 4.8. Light Sheet Microscopy

Whole organ imaging was performed with a custom LSM [33]. The light sheet was generated in digital scanning mode using a galvanometric mirror (6220H, Cambridge Technology, Bedford, MA, USA); confocal detection was achieved by synchronizing the galvo scanner with the line read-out of the sCMOS camera (Orca Flash4.0, Hamamatsu Photonics, Shizuoka, Japan). The laser light was provided by a diode laser (Cobolt, HÜBNER Photonics GmbH, Germany), and an acousto-optic tunable filter (AOTFnC-400.650-TN, AA Opto-Electronic, France) was used to adjust laser intensity. The excitation objective was a 10×, 0.3 NA Plan Fluor from Nikon, while the detection objective was a 10×, 0.6 NA Plan Apochromat from Olympus. The whole brain sample was recorded using a cuvette containing 40% TDE/PBS. The cuvette was mounted on a motorized x-, y-, z-, -stage (M-122.2DD and M-116.DG, Physik Instrumente, Karlsruhe, Germany), which allowed free 3D motion and rotation. Stacks were acquired with a z-step of approximately 3 µm and a xy resolution resulting from the setup configuration of 0.65 µm, with a field of view of 1.3 mm × 1.3 mm. The microscope was controlled via custom-written LabVIEW code (National Instruments, Austin, TX, USA), which coordinated the galvo scanners, the rolling shutter, and the stack acquisition.

### 4.9. Two Photon Fluorescence Microscope 

Slice imaging has been performed with a custom-made TPFM that enables mesoscopic reconstructions of cleared samples as performed in Costantini et al., 2021 [35]. A mode-locked Ti:Sapphire laser (Chameleon, 120 fs pulse width, 80 MHz repetition rate, Coherent, Santa Clara, CA, USA) operating at 800 nm was coupled into a custom-made scanning system based on a pair of galvanometric mirrors (LSKGG4/M, Thorlabs, Newton, NJ, USA). The laser was focused onto the specimen by a refractive index tunable 25× objective lens (LD LCI Plan-Apochromat 25×/0.8 290 Imm Corr DIC M27, Zeiss, Aalen, Germany). The system was equipped with a closed-loop XY stage (U780 PILine R 291 XY Stage System, 135 × 85 mm travel range, Physik Instrumente, Germany) for radial displacements of the sample and with a closed-loop piezoelectric stage (ND72Z2LAQ PIFOC objective scanning system, 2 mm travel range, Physik Instrumente, Karlsruhe, Germany) for the displacement of the objective along the z-axis. The fluorescence signal was collected by two independent GaAsP photomultiplier modules (H7422, Hamamatsu Photonics, Shizuoka, Japan). Emission filters of (440 ± 40) nm and (609 ± 54) nm were used to detect the signal, respectively, for DAPI and Eosin Y. The instrument was controlled by a custom software, written in LabView (National Instruments, USA), able to acquire a whole sample by performing z-stack imaging (depth = (500 ± 100) µm) of adjacent regions with an overlap of 40 µm and a voxel size of (0.88 × 0.88 × 2) µm^3^. The acquisition was performed with a dwell time of 500 µs, and the resulting 512 × 512 px images were saved as TIFF files.

### 4.10. Stitching and Pseudocoloring

High-resolution microscopy techniques such as LSM and TPFM, when used to image large biological samples, routinely produce big datasets that can easily reach several TB in size when dealing with whole organs, such as a whole mouse brain. Furthermore, due to their limited field of view, the images acquired by these instruments come in the form of a mosaic of 3D tiles that require volumetric stitching in order to reconstruct the whole imaged volume. For these reasons, early image post-processing is a specific challenge of these microscopy techniques that requires dedicated approaches and specialized tools. For 3D stitching, we used ZetaStitcher (https://github.com/lens-biophotonics/ZetaStitcher/, accessed on 1 March 2023), a custom Python tool specifically designed to be able to efficiently handle large datasets such as those produced in light-sheet microscopy [56]. Among the most important features of this package are its ability to perform a sampling along the tile depth to greatly reduce the amount of time and data needed to compute the alignment and an Application Programming Interface (API) that can be used to programmatically access the fused volume in a virtual fashion. After stitching, pseudo-coloring was applied to the reconstructed volumes to obtain a final image in the typical H&E colors. The pseudo-coloring step is simply a transformation of the color space, after normalization, according to the formulas provided by Torres et al. [34], where the channels for H and for E are remapped to RGB values:$$R=10^{-\left(0.644H+0.093E\right)}\;G=10^{-\left(0.717H+0.954E\right)}\;B=10^{-\left(0.267H+0.283E\right)}$$

## Figures and Tables

**Figure 1 ijms-24-06747-f001:**
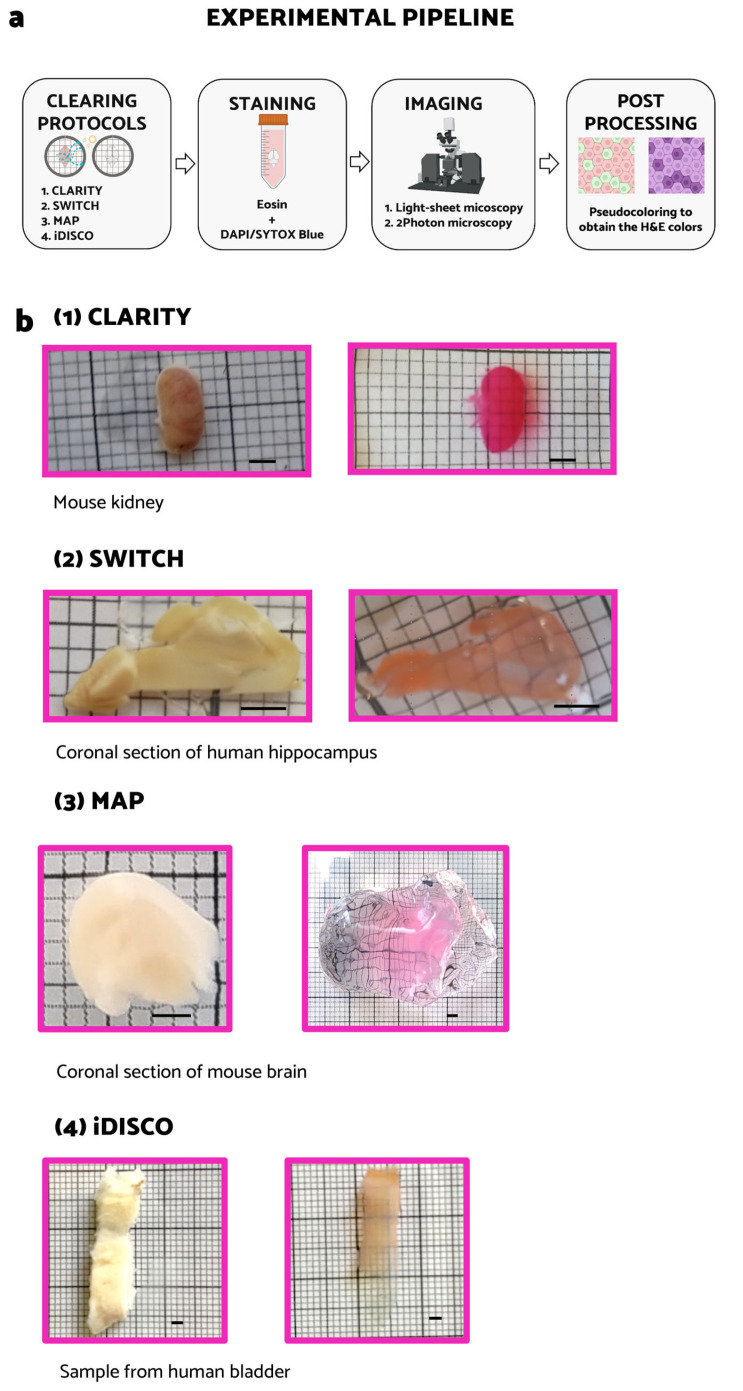
Tissue transformation and experimental pipeline. (**a**) Graphical abstract of the developed experimental pipeline which is composed of clearing, staining, imaging, and post-processing (image created with BioRender.com). (**b**) Comparison of clearing techniques applied to different biological samples before and after staining. (1) Mouse kidney before (left) and after (right) hydrogel-tissue transformation with CLARITY technique, staining, and thiodiethanol (TDE) refractive index (RI) matching. (2) Coronal section of human hippocampus before (left) and after (right) tissue transformation with SWITCH technique, staining, and TDE RI matching. (3) Coronal section of mouse brain before (left) and after (right) hydrogel-tissue transformation, staining, and expansion using MAP protocol. (4) Human bladder before (left) and after (right) clearing using iDISCO technique.

**Figure 2 ijms-24-06747-f002:**
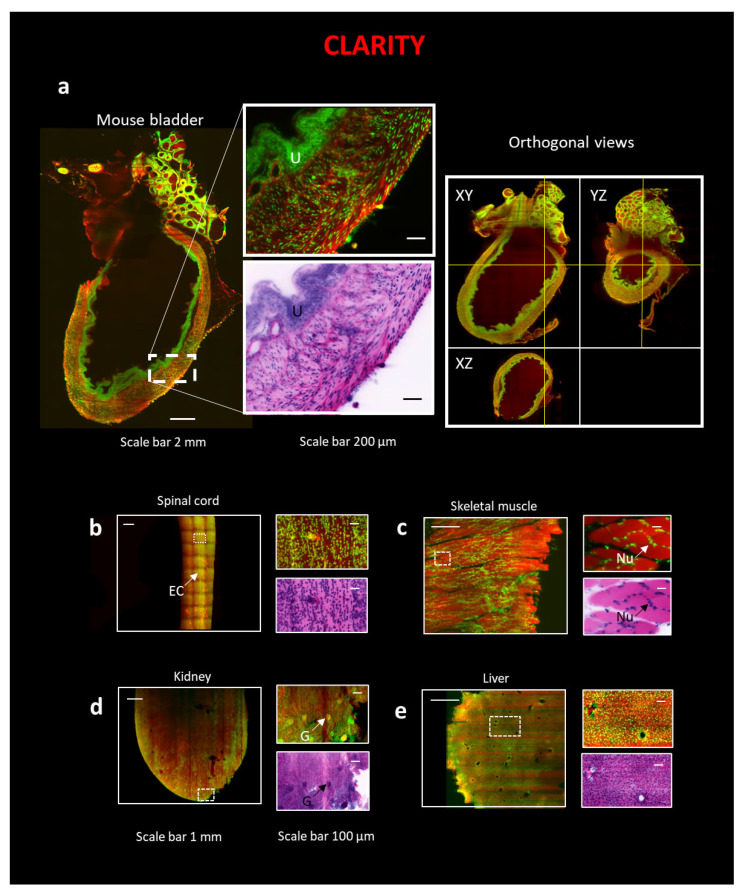
3D reconstruction of intact organs using the combination of CLARITY technique and LSM. (Left) All organs in the figure are stained with Eosin and SYTOX Blue. (Right) On the top of all images, a zoom-in on a particular region of interest (ROI) is represented, while on the bottom, the same ROI is converted with traditional H&E pseudo-coloration. The following organs are: (**a**) Mouse bladder, scale bar 2 mm, ROI, scale bar 200 µm. Orthogonal views of the intact mouse bladder. U—urothelium. (**b**) A portion of a slice of a mouse spinal cord, scale bar 1 mm, ROI, scale bar 100 µm; EC—Ependymal canal. (**c**) A portion of a slice of a mouse skeletal muscle, scale bar 1 mm, ROI, scale bar 100 µm. Nu—Nuclei. (**d**) A portion of a slice of a mouse kidney, scale bar 1 mm, ROI, scale bar 100 µm. G—Glomeruli. (**e**) Portion of a slice of a mouse liver, scale bar 1 mm, ROI, scale bar 100 µm.

**Figure 3 ijms-24-06747-f003:**
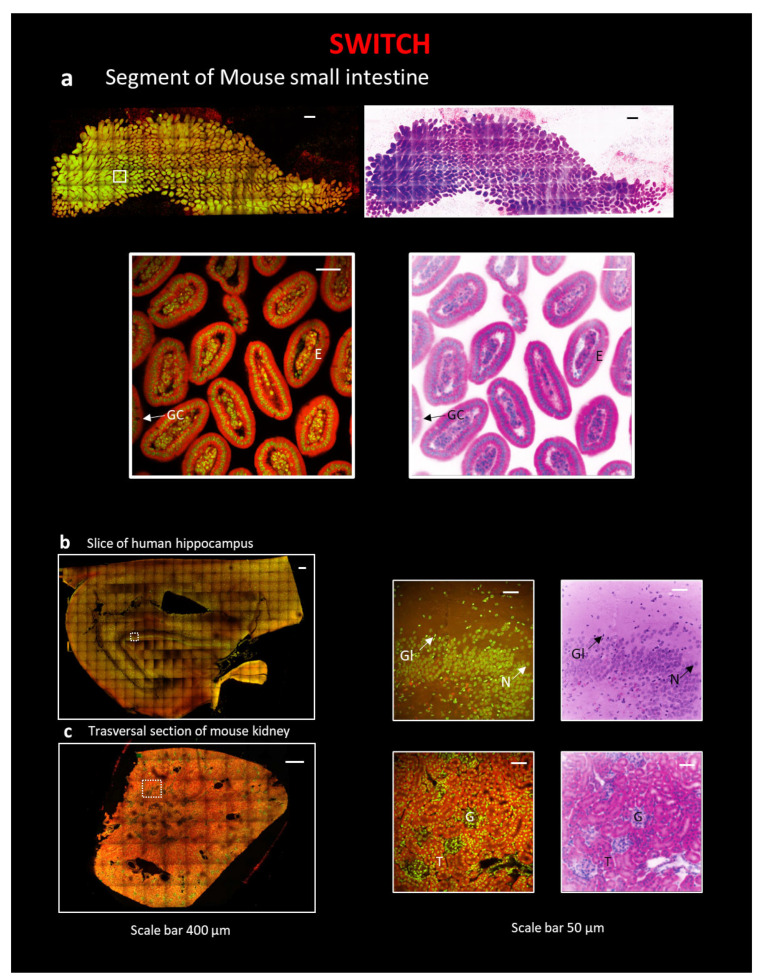
Reconstruction of tissue sections using the combination of SWITCH clearing protocol and TPFM. All sections in the figure are stained with Eosin and DAPI. Next, and under all images of the entire section, a zoom-in on a particular region of interest (ROI) is represented with traditional H&E fluorescent staining and pseudo-coloration. The following sections are: (**a**) A segment of mouse small intestine, scale 400 µm, ROI, scale bar 40 µm; E—columnar epithelium; GC—goblet cells. (**b**) A slice of human hippocampus, scale 400 µm, ROI, scale bar 40 µm. N—Neuron; Gl—Glial cell. (**c**) A transversal section of mouse kidney, scale 400 µm, ROI, scale bar 40 µm; G—Glomerulus; T—Tubule.

**Figure 4 ijms-24-06747-f004:**
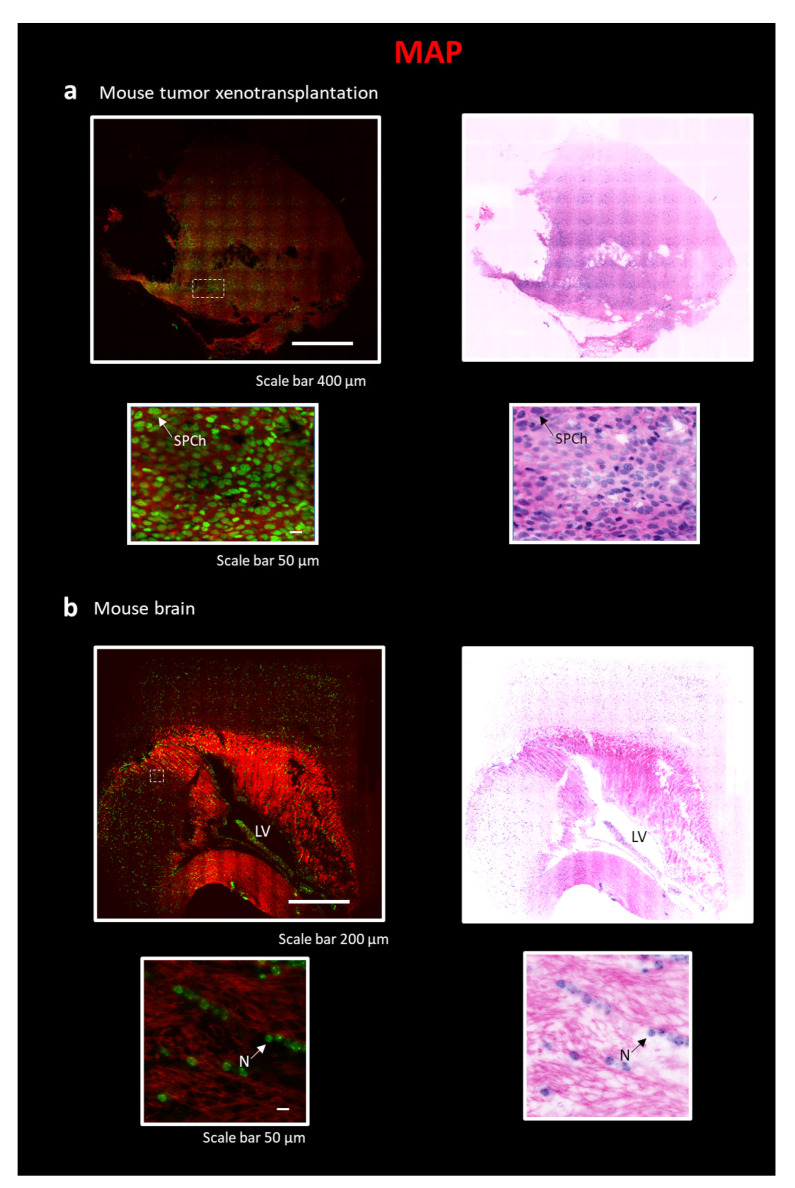
Reconstruction of tissue sections using the combination of MAP technique and LSM. All sections in the figure are stained with Eosin and SYTOX Blue, as described in Figure 2. On the top, entire sections are shown with traditional H&E fluorescent staining and pseudo-coloration. On the bottom, a particular region of interest (ROI) is represented. The following sections are: (**a**) Portion of a tumor xenotransplantation, scale 400 µm, ROI, scale bar 50 µm. SPCh—“salt and pepper chromatin”. (**b**) Slice of a mouse brain, scale 200 µm, ROI, scale bar 50 µm. LV—lateral ventricle; N—Neuron.

**Figure 5 ijms-24-06747-f005:**
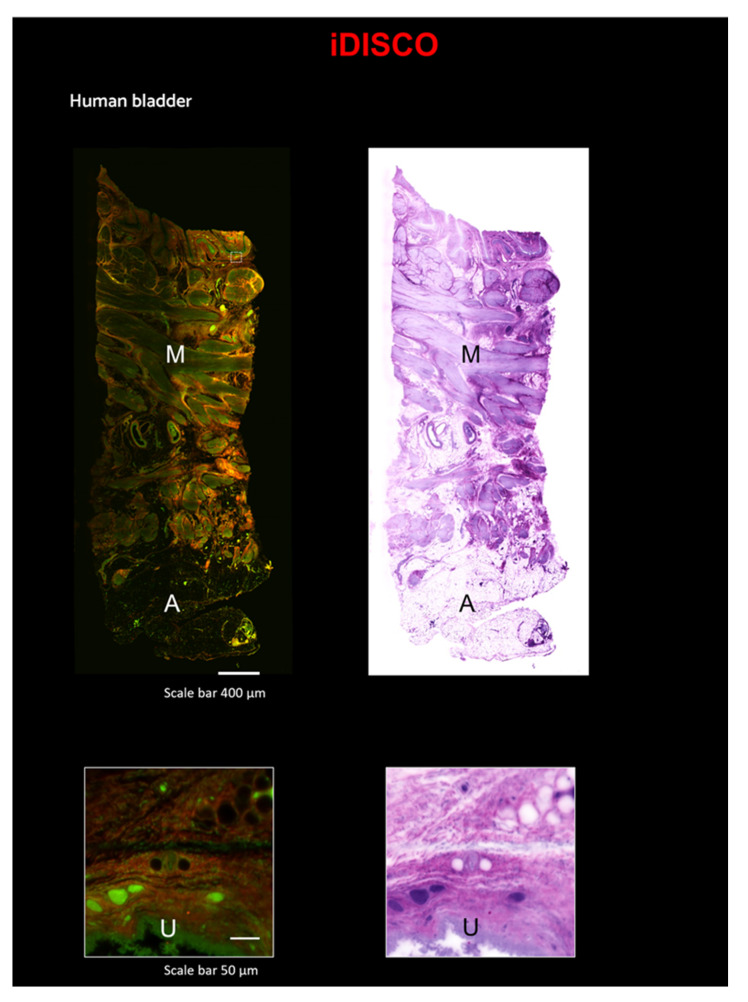
Reconstruction of a portion of the human bladder using a combination of the iDISCO technique and LSM. Top Left: Reconstruction of a portion of the human bladder stained with Eosin and SYTOX Blue, scale 400 µm. Top right: the same area converted with traditional H&E pseudo-coloration, scale 400 µm. Bottom left: Zoom-in on a particular ROI with Eosin and SYTOX Blue staining, scale bar 50 µm. Bottom right: Zoom-in on the same ROI converted with traditional H&E pseudo-coloration, scale bar 50 µm; A—Adipose tissue; M—Muscularis propria; U—Urothelium.

**Table 1 ijms-24-06747-t001:** Comparison of the different techniques used in this work (+ Fair; ++ Good; +++ Excellent). References are listed for the conditions used in different works (where present, they all confirm our findings).

	Human Tissue	Mouse Tissue	TPFM	LSM	Large Scale	Small Details
CLARITY	+ [15]	+++	+	+++ [15]	+++	+
SWITCH	+++	+++	+++	++	+	+++
MAP	+++	+++	++	+++	+	+++
iDISCO	+++ [19,21,23]	+++	++	+++ [21,23]	+++	++

## Data Availability

The data presented in this study are available on request from the corresponding author.

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
