# Peer review of "A Guide to Perform 3D Histology of Biological Tissues with Fluorescence Microscopy"

_ijms, 2023, doi:10.3390/ijms24076747_

Round 1

Reviewer 1 Report

The manuscript "A guide to perform 3D histology of biological tissues with fluorescence microscopy” describes the use of different methods to perform tissue clearing and fluorescence microscopy for three-dimensional histology applications. Four clearing protocols are applied to different tissues and data are compared and discussed. The topic is certainly of high and current interest and contributes to extend the current knowledge on novel tools for image analysis in research and pathology. 

Some minor points are listed below: 

Page 5 : this reviewer suggests replacing “that observed with classical H&E” with “those observed with classical H&E"

Page 10: Legend to figure 4: this reviewer suggests replacing “..SYTOX Blue as Fig.2 ” with “...SYTOX Blue as described in Fig.2”

Page 11: Figure 5 is missing

Please check Materials and Methods: some abbreviations are missing (DI, GA), names of manufacturers should have the same style (Sigma Aldrich, sigma or sigma Aldricht, Italy)

Reviewer 2 Report

No further comments.

Author Response

We thank the editor and reviewers for their thorough and careful evaluation of our manuscript. We have found all the comments very useful and have taken advantage of them to make our work stronger and more complete. In particular, following the suggestions provided by the reviewers, we checked for minor spelling errors, we added figure 5 to the current version of the manuscript and improved the “Materials and Methods” section. In addition, we added the “Literature Review” section, with more recent references on the use of clearing techniques, we added a paragraph in the “Discussion” section where we mention and discuss alternative techniques, we improved the “Result section 2.2” and the “Discussion" by adding the main limitations of each technique. We performed in-track changes the revisions made to the manuscript.

Reviewer 3 Report

1. The authors should provide the main contributions of this article at end of the Introduction section.

2. The authors should add the Literature Review section in this article.

3.    Hyperspectral and multispectral systems also play an important role in disease diagnosis. For example, “Smartphone imaging spectrometer for egg/meat freshness monitoring” and “Open-source mobile multispectral imaging system and its applications in biological sample sensing”, it is suggested that hyperspectral and multispectral systems should be discussed.

4. Why do the authors use SWITCH to treat a ~ 500 µm-thick slice of a human hippocampus in section 2.2 ? Please add the explanation in detail.

5. The authors should compare their work with the recent literature?

6. The authors should add the limitations of the proposed method that motivated the current research. 

7. The authors should cite the references in the recent five years.
